# Coexistence of Common Pathologies of the Cardiovascular System in a Patient with Pain in the Right Lower Limb

**DOI:** 10.3390/diagnostics11010056

**Published:** 2021-01-02

**Authors:** Paweł Gać, Martyna Hajac, Piotr Macek, Rafał Poręba

**Affiliations:** 1Centre for Diagnostic Imaging, 4th Military Hospital, Weigla 5, PL 50-981 Wroclaw, Poland; 2Department of Hygiene, Wroclaw Medical University, Mikulicza-Radeckiego 7, PL 50-368 Wroclaw, Poland; martyna.glosna@student.umed.wroc.pl; 3Department of Internal Medicine, Occupational Diseases and Hypertension, Wroclaw Medical University, Borowska 213, PL 50-556 Wroclaw, Poland; piotr.macek@umed.wroc.pl (P.M.); rafal.poreba@umed.wroc.pl (R.P.)

**Keywords:** aortic dissection, aortic aneurysm, computed tomography angiography, deep vein thrombosis, peripheral arterial atherosclerosis, pulmonary embolism

## Abstract

Deep vein thrombosis and pulmonary embolism, aortic aneurysm and aortic dissection, as well as peripheral arterial atherosclerosis, are frequently diagnosed vascular disorders. In this paper, the authors present the case of coexistence of common pathologies of the cardiovascular system in a patient with pain in the right lower limb. The presented images provide a didactically valuable overview of serious cardiovascular pathologies. This article highlights the value of computed tomography angiography in diagnosis of cardiovascular life-threatening conditions, especially as a result of proper medical interview and physical examination.

A 68-year-old man was transferred to the vascular emergency unit from a regional hospital where he was admitted due to pain in the right lower limb persisting for 3 days.

The anamnesis showed a history of stroke with aphasia, type 2 diabetes, persistent atrial fibrillation, chronic obstructive pulmonary disease (COPD) and osteoarthritis of the joints and spine. The patient denied taking any medications permanently and at that point described himself as an ex-smoker. On admission, the patient reported pain in the right lower limb and mild dyspnea.

Laboratory tests showed negative nasopharyngeal swab for SARS-CoV-2 (negative N2 gene for SARS-CoV-2 and E gene for the *Betacoronaviridae* family using the qRT-PCR method), slightly increased WBCs (11.6 × 103 µL), high CRP level (53.7 mg/L), marginally reduced sodium level (134 mmol/L) and significantly elevated D-dimer level (29.36 µg/mL).

The ultrasound examination found deep vein thrombosis in the right lower limb.

Owing to clinically suspected pulmonary embolism (high D-dimer concentration and presence of dyspnea), computed tomography angiography (CTA) of pulmonary arteries was performed. CTA showed dilation of the right and left pulmonary arteries with a borderline width of the pulmonary trunk (pulmonary trunk diameter about 34 mm, right pulmonary artery diameter about 27 mm, left pulmonary artery diameter about 27 mm) (Figure 1a). Distal section of the left pulmonary artery revealed a parietal irregular filling defect suggesting presence of embolic material (Figure 1b). No other embolic filling defects of arterial vessels were found. The right ventricle was normal-sized (right to left ventricular diameter ratio was 0.62); no angiographic signs of right heart failure were observed (no signs of contrast reflux into inferior vena cava (IVC)). Based on this CTA, diagnosis of pulmonary embolism was suggested. Additionally, CTA showed aortic dilation to about 6.0 cm in the ascending section and non-uniform aortic density suggesting acute aortic syndrome (Figure 1c).

On the basis of the aortic image on CTA of the pulmonary arteries and life-threatening condition, the decision was made to supplement diagnostics with CTA of the thoracic-abdominal aorta. CTA of the aorta showed aneurysm of the ascending aorta, aortic arch and proximal part of the descending aorta (maximum diameters of 6.0 cm, 3.9 cm and 4.9 cm, respectively) (Figure 1d). From the level of the left subclavian artery (LSA) origin over the further part of the thoracic aorta and proximal abdominal aorta, to the level of about 2.5 cm below the renal arteries origin, the vessel periphery revealed the presence of wide hypodense structure with thickness up to 1/2 diameter of the aortic lumen; in the proximal section, the structure was located medially and posteriorly, then left-sided and posteriorly, and in the distal section it was located both left-sided and anteriorly (Figure 1e). Regarding differentiation, the CTA image of the aorta could correspond to aortic aneurysm with massive parietal thrombi/aortic aneurysm with chronic dissection and clotting of the supposedly dissected canal. Moreover, CTA revealed angular (Gothic) aortic arch (Figure 1f), short-segment stenosis of the proximal section of the superior mesenteric artery by 50–70% of the lumen (Figure 1g), occlusion of the proximal section of the right superficial femoral artery (Figure 1h) and filling defects in the veins of the right iliac axis that is most probably a thrombotic material (Figure 1i).

Due to development of symptoms of acute ischemia in the right lower limb, the patient was qualified for surgical treatment. Embolectomy of the common femoral artery, deep femoral artery, superficial femoral artery and popliteal artery was performed with improvement in the blood supply to the operated limb. Antithrombotic treatment was introduced. The patient was qualified for planned cardiac surgery of the ascending aortic aneurysm.

Deep vein thrombosis and pulmonary embolism, aortic aneurysm and aortic dissection, as well as peripheral arterial atherosclerosis are frequently diagnosed vascular disorders, although they do not tend to coexist [1,2,3,4]. The separate prognoses for each of these diseases are serious [1,2,3,4]. The co-occurrence of various diseases of the cardiovascular system is occasionally observed, more often in the elderly [5]. Due to this fact, increased risk of deep vein thrombosis and pulmonary thromboembolism in patients with aortic aneurysm recently became an area of investigations [6].

The presented images provide a didactically valuable overview of serious cardiovascular pathologies and based on literature review no similar case reports of such complex pathologies of cardiovascular diseases were described. This article highlights the value of CTA in diagnosis of cardiovascular life-threatening conditions, especially as a result of proper medical interview and physical examination (which revealed risk factors-persistent atrial fibrillation, dyspnea and limb pain). The partnership of radiologist and emergency physician in such cases is essential.

## Figures and Tables

**Figure 1 diagnostics-11-00056-f001:**
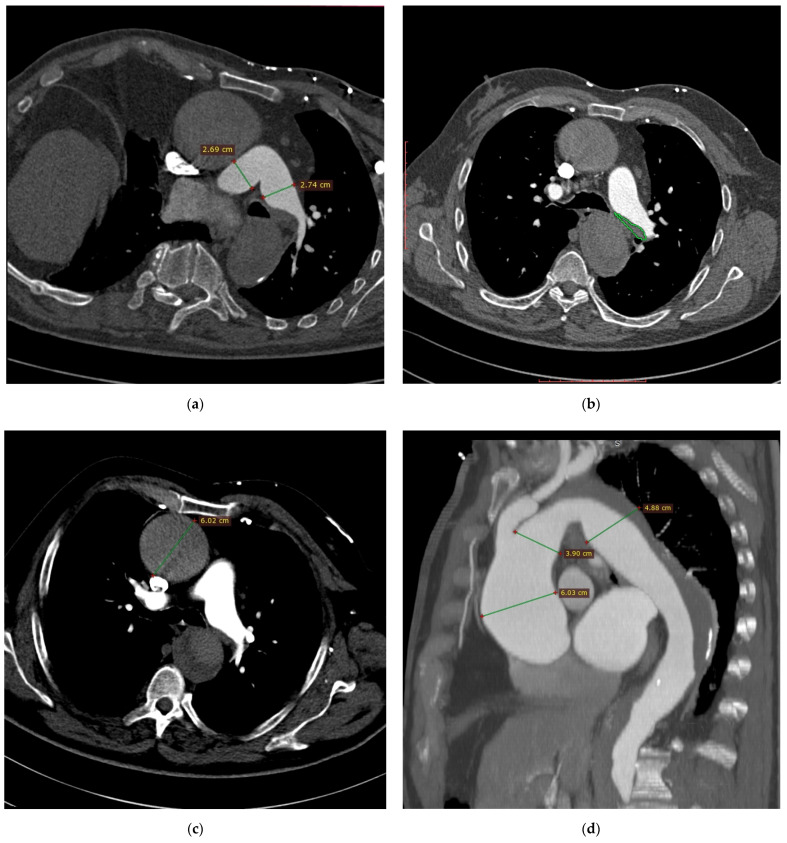
Computed tomography angiography images: (**a**) dilation of the right and left pulmonary arteries with a borderline width of the pulmonary trunk; (**b**) left pulmonary artery filling defect suggesting presence of embolic material; (**c**) non-uniform aortic density suggesting acute aortic syndrome; (**d**) aneurysm of the ascending aorta, aortic arch and proximal part of the descending aorta; (**e**) aortic aneurysm with massive parietal thrombi/aortic aneurysm with chronic dissection and clotting of the supposedly dissected canal; (**f**) angular (Gothic) aortic arch; (**g**) stenosis of the proximal section of the superior mesenteric artery; (**h**) occlusion of the proximal section of the right superficial femoral artery; (**i**) filling defects in the veins of the right iliac axis that is most probably a thrombotic material.

## Data Availability

No new data were created or analyzed in this study. Data sharing is not applicable to this article.

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
