# Peer review of "Coexistence of Common Pathologies of the Cardiovascular System in a Patient with Pain in the Right Lower Limb"

_diagnostics, 2021, doi:10.3390/diagnostics11010056_

Round 1

Reviewer 1 Report

The manuscript entitled "Coexistence of common pathologies of the cardiovascular system in a patient with pain in the right lower limb" reports a case report of a patient arrived at the hospital with a serious clinical situation. The paper is simple, clear, and descriptive. I think that it can be published.

I think that the manuscript entitled "Coexistence of common pathologies of the cardiovascular system in a patient with pain in the right lower limb" should be accepted for the publication in the present form.

Author Response

Dear Reviewer,

Thank you for careful and thorough reading of this manuscript and for the thoughtful comments. We are incredibly grateful for the review.

Comment of Reviewer: The manuscript entitled "Coexistence of common pathologies of the cardiovascular system in a patient with pain in the right lower limb" reports a case report of a patient arrived at the hospital with a serious clinical situation. The paper is simple, clear, and descriptive. I think that it can be published.

I think that the manuscript entitled "Coexistence of common pathologies of the cardiovascular system in a patient with pain in the right lower limb" should be accepted for the publication in the present form.

Authors reply: Again, thank you for the positive review of our article.

Best regards,

Authors

Reviewer 2 Report

This is an interesting case for everyday practice. The manuscript shows good pictures records. For the practitioner, this case illustration is a teachable example. The authors gave a good presentation! The manuscript can be accepted for publication in its present form.

Author Response

Dear Reviewer,

Thank you for careful and thorough reading of this manuscript and for the thoughtful comments. We are incredibly grateful for the review.

Comment of Reviewer: This is an interesting case for everyday practice. The manuscript shows good pictures records. For the practitioner, this case illustration is a teachable example. The authors gave a good presentation! The manuscript can be accepted for publication in its present form.

Authors reply: Thank you again for your positive review of our article.

Best regards,

Authors

Reviewer 3 Report

Thank you for the review opportunity. Article is well written. Although no specific changes are required for this manuscript, this paper is a general presentation of a patients with multiple comorbidities that are incidentally diagnosed after CT scan and adequate investigation, and adds very few data to the existing literature.

Author Response

Dear Reviewer,

Thank you for careful and thorough reading of this manuscript and for the thoughtful comments. We are incredibly grateful for the review.

Comment of Reviewer: Thank you for the review opportunity. Article is well written. Although no specific changes are required for this manuscript, this paper is a general presentation of a patients with multiple comorbidities that are incidentally diagnosed after CT scan and adequate investigation and adds very few data to the existing literature.

Authors reply: Thanks for your comment. In the summary of the description, we indicate our motivation for the publication of this case study: "The presented images provide a didactically valuable overview of serious cardiovascular pathologies and based on literature review no similar case reports of such complex pathologies of cardiovascular diseases were described. This article highlights the value of CTA in diagnosis of cardiovascular life-threatening conditions, especially as a result of proper medical interview and physical examination (which revealed risk factors - persistent atrial fibrillation, dyspnea and limb pain). Partnership of radiologist and emergency physician in such cases is essential. "

Best regards,

Authors

This manuscript is a resubmission of an earlier submission. The following is a list of the peer review reports and author responses from that submission.

Round 1

Reviewer 1 Report

The manuscript entitled "Coexistence of common pathologies of the cardiovascular system in a patient with pain in the right lower limb" reports a case report of a patient arrived at the hospital with a serious clinical situation. The paper is simple, clear, and descriptive. I think that it can be published.

Author Response

Dear Reviewer,

Thank you for careful and thorough reading of this manuscript and for the thoughtful comments. We are incredibly grateful for the review.

Comment of Reviewer: The manuscript entitled "Coexistence of common pathologies of the cardiovascular system in a patient with pain in the right lower limb" reports a case report of a patient arrived at the hospital with a serious clinical situation. The paper is simple, clear, and descriptive. I think that it can be published.

Changes carried out in the paper: The reviewer does not indicate the need to amend the article. Thanks again for the positive review of the article.

Best regards,

Authors

Reviewer 2 Report

The authors present a case of a 68 year old male patient with extensive vascular disease. Indeed, the patient has a remarkable myriad of vascular pathologies, but this is not that rare. This case report would be strengthened if the authors could check English grammar and spelling. Please provide more info on the background of this patient. E.g. did he smoke? What are the CV risk factors? Could the authors comment on the specific anatomy of the aortic arch (Gothic arch)? A 3D reconstruction of the aorta would also add some additional info.

Author Response

Dear Reviewer,

Thank you for careful and thorough reading of this manuscript and for the thoughtful comments. We are incredibly grateful for the review.

Comment of Reviewer: The authors present a case of a 68-year-old male patient with extensive vascular disease. Indeed, the patient has a remarkable myriad of vascular pathologies, but this is not that rare.

Changes carried out in the paper: The conclusion of the short commentary to the case report has been corrected to the present wording: "The presented images provide a typical, didactically valuable overview of serious cardiovascular pathologies."

Comment of Reviewer: This case report would be strengthened if the authors could check English grammar and spelling.

Changes carried out in the paper: The following sentences have been corrected in terms of grammar and style:

- On admission, the patient reported, aside to pain in the right lower limb, slight dyspnea.

- Laboratory tests showed negative nasopharyngeal swab for SARS-CoV-2 (negative N2 gene for SARS-CoV-2 and E gene for the Betacoronaviridae family using the qRT-PCR method), slightly increased WBCs (11.6 x 103 µL), high CRP level (53.7 mg/L), slightly reduced sodium level (134 mmol/L) and significantly increased d-dimer level (29.36 µg/mL).

- The ultrasound examination found deep vein thrombosis in the right lower limb.

- Because of clinically suspected pulmonary embolism (high d-dimer concentration and a history of dyspnea), computed tomography angiography (CTA) of pulmonary arteries was performed.

- Right to left ventricular diameter ratio was 0.62.

- In summary, based on the CTA of the pulmonary arteries, diagnosis of pulmonary embolism was suggested.

- CTA of the aorta revealed aneurysm of the ascending aorta, aortic arch and proximal part of the descending aorta.

- From the level of the LSA origin over the further part of the thoracic aorta and proximal abdominal aorta, to the level of about 2.5 cm below the renal arteries origin, the vessel periphery revealed the presence of wide hypodense structure with thickness up to 1/2 diameter of the aortic lumen.

- Due to development of symptoms of acute ischemia in the right lower limb, the patient was qualified for surgical treatment.

Comment of Reviewer: Please provide more info on the background of this patient. E.g. did he smoke? What are the CV risk factors?

Changes carried out in the paper: The first version of the manuscript limited the clinical data on the patient due to journal requirements (the article was submitted to the "interesting images" section, not the "case report" section). Currently, the patient's characteristics have been supplemented with the data collected from the medical interview: "The anamnesis showed a history of stroke with aphasia, type 2 diabetes, persistent atrial fibrillation, chronic obstructive pulmonary disease (COPD) and osteoarthritis of the joints and spine. The patient denied taking medications permanently. He has declared to have smoked in the past and now describes himself as a non-smoker.”

Comment of Reviewer: Could the authors comment on the specific anatomy of the aortic arch (Gothic arch)? A 3D reconstruction of the aorta would also add some additional info.

Changes carried out in the paper: Thank you for identifying a clinically important variant of the aortic arch. Relevant information was added to the text: "Moreover, CTA revealed angular (Gothic) aortic arch, ...". Figure 1F has been added - 3D visualization of the aortic arch.

Best regards,

Authors

Round 2

Reviewer 2 Report

Thank you for your revision. 

The paper was improved, but I believe the English can be still be optimized. Also, in my opinion, there is no real novelty regarding this case, other than the combination of pathologies.